# Dissociative Adsorption of Hydrogen Molecules at Al$_2$O$_3$ Inclusions in Steels and Its Implications for Gaseous Hydrogen Embrittlement of Pipelines

Yinghao Sun and Frank Cheng *

Department of Mechanical & Manufacturing Engineering, University of Calgary, Calgary, AB T2N 1N4, Canada; yinghao.sun1@ucalgary.ca
* Correspondence: fcheng@ucalgary.ca

**Abstract:** Hydrogen embrittlement (HE) of steel pipelines in high-pressure gaseous environments is a potential threat to the pipeline integrity. The occurrence of gaseous HE is subjected to associative adsorption of hydrogen molecules (H$_2$) at specific "active sites", such as grain boundaries and dislocations on the steel surface, to generate hydrogen atoms (H). Non-metallic inclusions are another type of metallurgical defect potentially serving as "active sites" to cause the dissociative adsorption of H$_2$. Al$_2$O$_3$ is a common inclusion contained in pipeline steels. In this work, the dissociative adsorption of hydrogen at the $\alpha$-Al$_2$O$_3$(0001)/$\alpha$-Fe(111) interface on the Fe$(01\bar{1})$ plane was studied by density functional theory calculations. The impact of gas components of O$_2$ and CH$_4$ on the dissociative adsorption of hydrogen was determined. The occurrence of dissociative adsorption of hydrogen at the Al$_2$O$_3$ inclusion/Fe interface is favored under conditions relevant to pipeline operation. Thermodynamic feasibility was observed for Fe and O atoms, but not for Al atoms. H atoms can form more stable adsorption configurations on the Fe side of the interface, while it is less likely for H atoms to adsorb on the Al$_2$O$_3$ side. There is a greater tendency for the occurrence of dissociative adsorption of O$_2$ and CH$_4$ than of H$_2$, due to the more favorable energetics of the former. In particular, the dissociative adsorption of O$_2$ is preferential over that of CH$_4$. The Al-terminated interface exhibits a higher H binding energy compared to the O-terminated interface, indicating a preference for hydrogen accumulation at the Al-terminated interface.

**Keywords:** hydrogen gas; dissociative adsorption; pipelines; Al$_2$O$_3$ inclusion; density functional theory

## 1. Introduction

Hydrogen, as a clean energy carrier, has been acknowledged as a critical player in energy transition and pursuit of the 2050 net-zero target [1–4]. Pipelines offer an economical, effective, and efficient means to transport large-capacity hydrogen over vast distances [5–8]. However, steel pipelines are considered to be susceptible to hydrogen embrittlement (HE) in high-pressure gaseous hydrogen environments. The gaseous HE can compromise the structural integrity, causing pipeline failures [9]. The occurrence of gaseous HE in pipelines is subjected to the generation, adsorption, and absorption of H atoms into the steel [10–12]. Due to the size limitation, H$_2$ molecules cannot enter the steels. In gaseous environments, the generation of H atoms takes place through the dissociative adsorption of H$_2$ molecules at specific "active sites", such as low-index crystalline planes [13–16] and the grain boundaries [17,18] and dislocations [19] on the steel surface.

The distinct H–iron (Fe) atomic interaction in gaseous environments has been studied by various experimental techniques, such as scanning Kelvin probe force microscopy (SKPFM) [20], low-energy electron diffraction (LEED) [21], and electron energy-loss spectroscopy (EELS) [22]. Some crucial findings have been obtained, including H adsorption energies at specific crystalline planes [21], configurations of H adsorption [22,23], and the distribution of H atoms within the steels. Given the challenges in experimentation and the

limitations in understanding the detailed configurations and mechanisms of H–Fe bonding through hybridization, contemporary computational methods like density functional theory (DFT) present promising avenues for investigating H–Fe interactions in gaseous environments. It was determined that the H adsorption energies on low-index Fe crystalline planes, including Fe(100) [13], Fe(110) [24], and Fe(111) [25] planes, ranged between −0.4 eV and −0.6 eV. Upon H–Fe hybridization, electrons shift from the Fe atoms to the H atoms, resulting in electron consumption on the Fe atoms and electron accumulation on the H atoms. The charged H atoms repel each other and cause cleavage of H–H bonds. The generated H atoms exhibit local adsorption on the steel. Subsequently, the adsorbed H atoms have the ability to permeate into the subsurface of the Fe plane, becoming absorbed H atoms that predominantly occupy tetrahedral sites within the Fe lattice [14,15]. The H atoms tend to diffuse towards high-stress regions or become trapped at various metallurgical features, such as grain boundaries, dislocations, non-metallic inclusions, and secondary-phase particles. When the local concentration of H atoms surpasses a threshold value, cracks can be initiated, particularly when subjected to applied stress. Various mechanisms or models have been proposed to elucidate the phenomenon of hydrogen-induced cracking. The predominant mechanisms encompass hydrogen-enhanced decohesion (HEDE) and hydrogen-enhanced localized plasticity (HELP) [12,26].

In recent years, one of the central research focuses in the realm of gaseous HE of pipelines has been the dissociative adsorption of $H_2$ molecules at "active sites" on steels. Particularly, the high-angle grain boundary (HAGB) on the surface of steels can facilitate the occurrence of dissociative adsorption of hydrogen [18]. The H adsorption energy at the HAGB is more negative compared to the crystalline planes, promoting the localized accumulation of adsorbed H atoms in this region. In addition, the emergence of dislocations on the steel surface exhibits a preference for adsorbing dissociated H atoms over Fe lattice planes [19]. Dissociative adsorption of hydrogen was also investigated on oxide scales, commonly found on the pipe body [27–29]. It was demonstrated that nearly half of the sites on $Fe_2O_3$ oxide do not favor H adsorption, while the rest of the sites facilitate $H_2$ dissociation and tightly bind the generated H atoms. As a result, the oxide inhibits the permeation of H atoms into steels [30]. The partition function has been widely employed to incorporate the influences of environmental temperature and pressure into the dissociative adsorption of hydrogen [18,19,30–32]. In general, elevated pressure and low temperatures promote the occurrence of dissociative adsorption of hydrogen. Furthermore, the impurity gases within the fluid in pipelines will affect the dissociative adsorption of $H_2$ molecules [33,34]. Gases containing electronegative elemental oxygen, such as $O_2$, $H_2O$, and CO, compete with $H_2$ molecules and preferentially adsorb on the steel surface, inhibiting the generation of H atoms. However, methane ($CH_4$) does not show an apparent impact on the dissociative adsorption of hydrogen.

In addition to the atomic-scale metallurgical features like grain boundaries and dislocations, non-metallic inclusions contained in pipeline steels serve as effective H traps. The inclusions are usually introduced into steels during manufacturing, deoxidization, and desulfurization [35]. They can impact H-induced cracking of steels drastically [36]. It was found that mixed inclusions increased the susceptibility of steels to HE, but pure sulfide inclusions rarely caused the initiation of cracks [37,38]. The oxide inclusions enriched in Al, Ti, and Mn were associated with H-induced cracking [39,40], and few cracks were observed at Si-enriched inclusions [41]. Particularly, $Al_2O_3$ inclusions can initiate H-induced cracking by virtue of H atom enrichment at the $Al_2O_3$/steel interfaces [42,43]. Theoretical calculations yielded H binding energy, $E_{\text{binding}}^{\text{H}}$, at various inclusions, spanning a range from 0.13 eV to 0.29 eV. The $E_{\text{binding}}^{\text{H}}$ of the $Al_2O_3$ inclusion was 0.24 eV, suggesting a robust bonding between H atoms and the $Al_2O_3$ inclusion [37,44]. For comparison, the $E_{\text{binding}}^{\text{H}}$ at the lattice site was only about 0.08 eV [45]. There exist two distinct Fe–$Al_2O_3$ interfacial structures, namely, the Al-terminated interface and the O-terminated interface [46]. The O-terminated interface exhibits a higher bonding strength and electron concentration

compared to the Al-terminated interface. Both chemical bonding and orbital hybridization contribute to interfacial cohesion.

As of now, there has been limited work investigating the dissociative adsorption of $H_2$ molecules at inclusions in gaseous environments. This is a big gap to be filled in the steel HE area. The novelty of this work lies in its investigation of the dissociative adsorption of hydrogen at the $Al_2O_3$ inclusion/Fe interface within pipeline steels. The focus is on identifying preferential H adsorption sites, determining configurations, and quantifying the H adsorption energy at this specific interface. The electron shift pattern is defined through partial density of states (PDOS) analysis. The partition function was applied to correct the temperature/pressure conditions. The influence of impurity gases of $O_2$ and $CH_4$ in the fluid on the dissociative adsorption of hydrogen was studied. Moreover, the hydrogen trapping at the $Al_2O_3$ inclusion/Fe interface with various terminations was investigated. It is expected that this work will establish a knowledge base regarding the dissociative adsorption of $H_2$ molecules at $Al_2O_3$ inclusions in steels, along with the generation of H atoms in high-pressure gaseous environments.

## 2. Computational Methodology

### 2.1. Modeling of the α-Al₂O₃(0001)/α-Fe(111) Interface on the Fe$(01\bar{1})$ Crystalline Plane

The first step in model development in this work was to create an $Al_2O_3$/Fe interface. The most stable plane in the Fe lattice was selected as the base plane to create the interface [47]. The plane orientation and lattice mismatch were considered for modeling. α-$Al_2O_3$ is the most stable phase of various $Al_2O_3$ configurations [48]. The primary cell of α-$Al_2O_3$ and its (0001) plane, which was selected due to having the lowest surface energy, are shown in Figure 1a [49]. The primary cell of α-Fe and its (111) plane, which served as the base plane, are shown in Figure 1b. The α-Fe(111) plane was used to create the interface, since this plane exhibits similar interaxial angles to the α-$Al_2O_3$(0001) plane, i.e., α = β = 90° and γ = 120°. The 30° orientation angle was applied to minimize mismatch of the α-$Al_2O_3$(0001) and α-Fe(111) planes, as shown in Figure 1c,d. This orientation relationship was previously experimentally verified as the preferential choice [50]. The bonds within the α-$Al_2O_3$(0001)/α-Fe(111) interface were removed to clearly visualize the adsorption/trapping configurations of hydrogen and impurity gases. The lattice constant for the optimized BCC Fe is 2.8656 Å, which is consistent with the experimental data of $a$ = 2.866 Å [51]. The geometrically optimized lattice constants of α-$Al_2O_3$ include $a = b$ = 4.80 Å and $c$ = 13.12 Å, showing a good consistency with existing data [46,52]. The lattice constants of the modeled cells in Figure 1e,f include $a = b$ = 8.1751 Å and $c$ = 25.2278 Å. To calculate the interface distances, all atoms were fully relaxed after construction of the interface layers according to an existing model [46] with an Fe/Al oxide system. The steps were as follows: (i) Construct an interface layer model according to the existing interface data; a 10 Å vacuum slab was added and a supercell was created, (ii) Set all atoms in the model to be fully relaxed. (iii) Conduct geometry optimization along the Z direction to determine the interface distance. (iv) Measure the distance of the studied interface. The distances of the Al-terminated and the O-terminated interfaces were calculated as 1.74 Å and 1.29 Å, respectively. The two layers of the Fe cell from the bottom and the top three layers of $Al_2O_3$ were subsequently fixed for further optimization. A vacuum slab of 10 Å was added on the top of the $Al_2O_3$, eliminating the influence of periodic boundaries. Both the Al-terminated and the O-terminated interfaces were created and geometrically optimized, as shown in Figure 1e,f, where the boxes marked with dashed lines are the supercells applied in this study and the dashed lines indicate periodic boundaries.

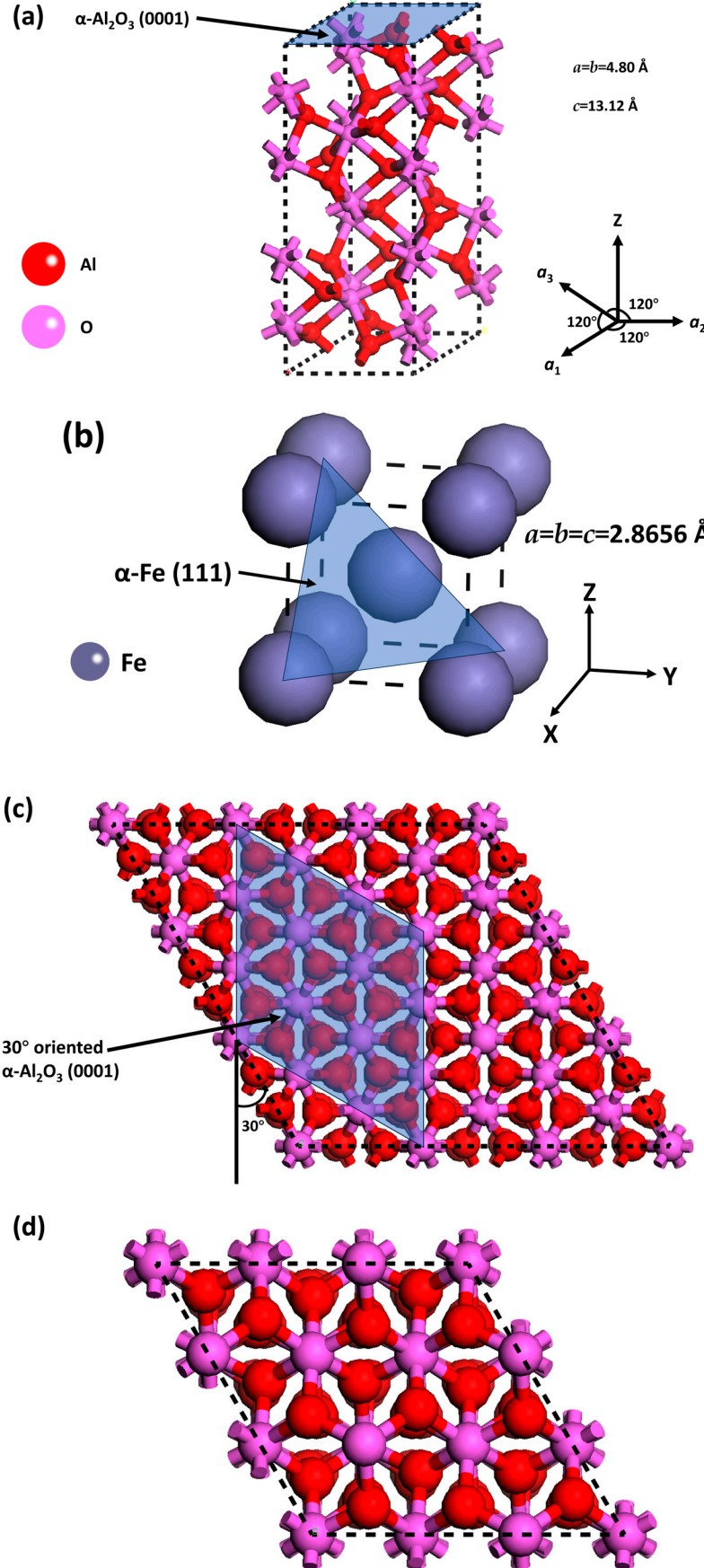

**Figure 1.** *Cont.*

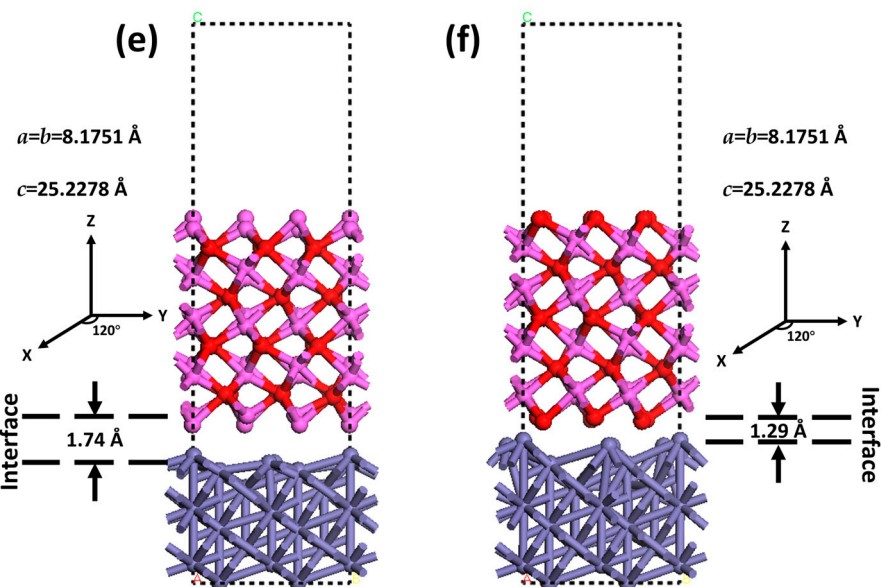

**Figure 1.** (**a**) Primary cell of $\alpha$-Al$_2$O$_3$ and its (0001) plane used in interface construction. (**b**) Primary cell of $\alpha$-Fe and its (111) plane used in interface construction. (**c**) Top view of the $\alpha$-Al$_2$O$_3$(0001) plane and its 30° oriented plane. (**d**) Top view of the 30° oriented $\alpha$-Al$_2$O$_3$(0001) plane used in this study. (**e**) The Al-terminated $\alpha$-Al$_2$O$_3$(0001)/$\alpha$-Fe(111) interface, and (**f**) the O-terminated $\alpha$-Al$_2$O$_3$(0001)/$\alpha$-Fe(111) interface. The tiny coordinate systems in (**a**–**d**) indicate the interaxial angles. The boxes marked with dashed lines in (**e**,**f**) are the supercell applied in this study, and the dashed lines indicate periodic boundaries. The solid lines represent the interatomic bonds, illustrating the lattice structure and orientations.

The $\alpha$-Fe contained in pipeline steels exhibits a body-centered cubic (BCC) structure [53]. The close-packed plane for BCC Fe belongs to the {110} family [54]. Compared with other low-index crystalline planes, such as Fe(100) and Fe(111) planes, the dissociative adsorption of H$_2$ molecules on the Fe(110) plane can occur over a wide temperature/pressure range, along with the greatest H adsorption energy [21,31]. The Fe(110) plane is thus the preferential site for the occurrence of dissociative adsorption of hydrogen, leading to the generation of H atoms. The Fe lattice plane used in this work was the Fe{110} family. The Fe$\left(01\bar{1}\right)$ plane was selected to coordinate with the modeled $\alpha$-Al$_2$O$_3$(0001)/$\alpha$-Fe(111) interface. Therefore, the $\alpha$-Al$_2$O$_3$(0001)/$\alpha$-Fe(111) interface was created on the Fe$\left(01\bar{1}\right)$ plane, as illustrated in Figure 2. The 8 Å and 15 Å vacuum slabs were added on the side and the top of the structure, respectively. It has been proposed that four layers in a BCC Fe slab can make the surface energy consistent [31,55,56]. For $\alpha$-Al$_2$O$_3$, it was verified that six layers can make the surface energy achieve consistency [57]. Therefore, six layers of Al$_2$O$_3$ inclusions and four layers of Fe matrix were involved in the plane modeling. Given the periodic features of metals and their oxides, the bulk region rarely shows changes in the interlayer space and configuration. Thus, the bottom layers in the slab can be fixed, while atoms that are near the surface and adsorbent are allowed to relax [54]. It is common to fix the bottom two layers and relax the top two layers for BCC Fe. The first interlayer space, i.e., 2.04 Å, is close to the bulk value, i.e., 2.03 Å [58]. Moreover, it was experimentally determined that the top two or three layers of $\alpha$-Al$_2$O$_3$ lost their periodic structure while the other layers retained their periodicity [59,60]. Thus, the three bottom layers of the Al$_2$O$_3$ inclusions and two layers in the bottom of the Fe matrix were fixed, as indicated in Figure 2b. Moreover, three layers at the sides were also fixed (Figure 2c). Only the plane with the Al-terminated Al$_2$O$_3$ was used in this work, as previous works have confirmed that the Al-terminated interface exhibits lower adhesion and is prone to dissociative adsorption of hydrogen and crack initiation [46].

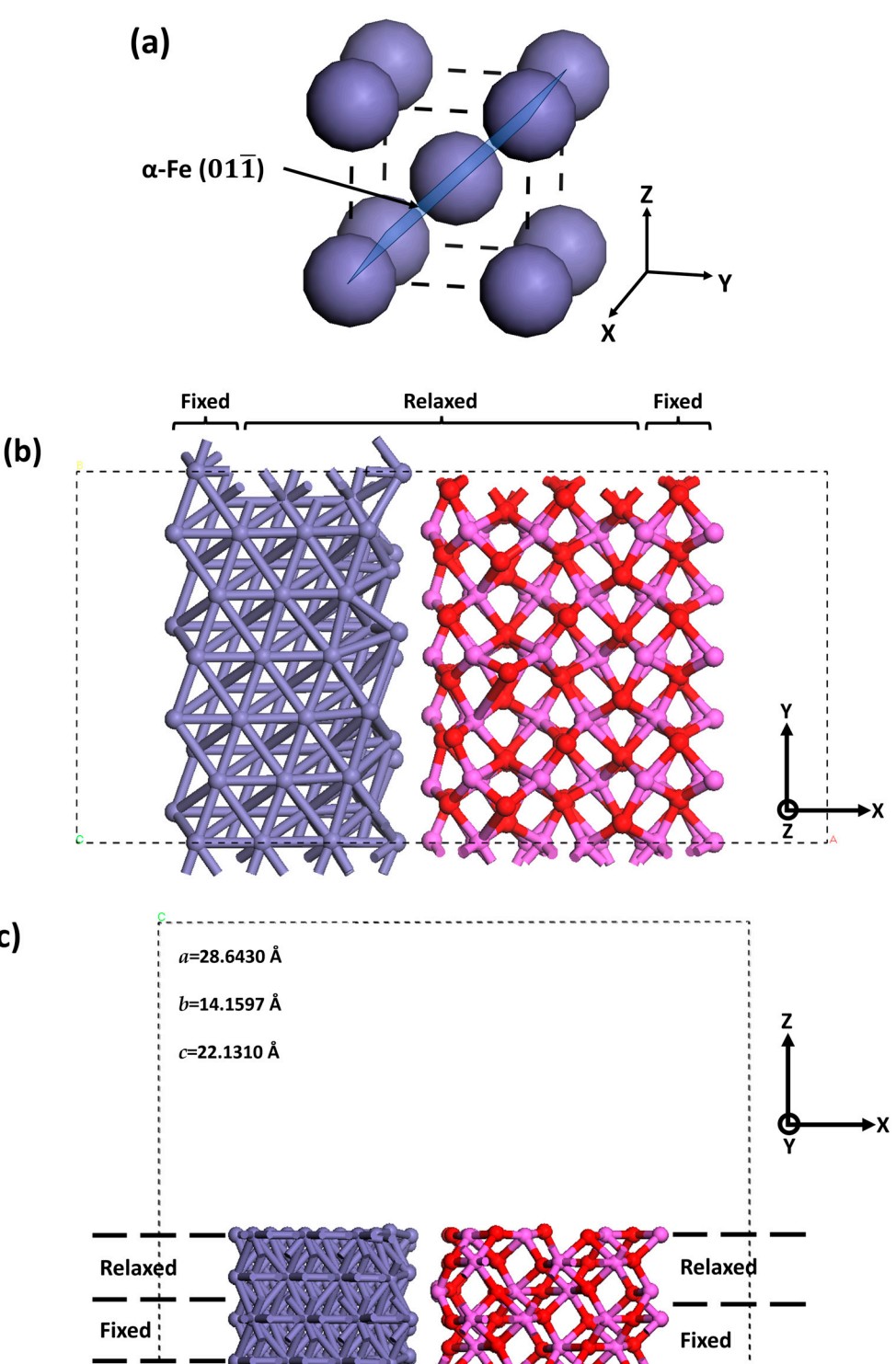

**Figure 2.** (**a**) Primary cell of $\alpha$-Fe and its $(01\bar{1})$ plane used in surface cleavage. (**b**) Top view of the geometrically optimized $\alpha$-Al$_2$O$_3$(0001)/$\alpha$-Fe(111) interface on the Fe$(01\bar{1})$ plane. (**c**) Side view of the geometrically optimized $\alpha$-Al$_2$O$_3$(0001)/$\alpha$-Fe(111) interface on the Fe$(01\bar{1})$ plane. The boxes marked with dashed lines represent the supercell applied in this study. The dashed lines indicate periodic boundaries. Purple: Fe atoms. Pink: Al atoms. Red: O atoms.

## 2.2. Change in Free Energy for Dissociative Adsorption of Hydrogen

Computational work for the dissociative adsorption of hydrogen is commonly conducted under ideal conditions, typically in a vacuum at 0 K, with the influence of tempera-

ture and pressure on the motion and adsorption of atoms and molecules often neglected. The partition function was applied in this study to correct the motional contribution of gaseous hydrogen in the change in free energy associated with the dissociative adsorption of hydrogen. This correction, considering the impact of environmental factors such as temperature and pressure, has been widely used in the modeling of gas–solid interactions [30–32]. The change in free energy is a criterion to determine the thermodynamics of a reaction. A negative change in free energy indicates that the reaction is spontaneous under given conditions. The dissociative adsorption of $H_2$ molecules at the $\alpha$-$Al_2O_3$(0001)/$\alpha$-Fe(111) interface on the Fe$(01\bar{1})$ plane, leading to the generation of H atoms, is described as follows:

$$\frac{1}{2}nH_2(gas) + \left[Fe(01\bar{1}) + \alpha\text{-}Al_2O_3(0001)/\alpha\text{-}Fe(111)\right] \rightarrow \left[Fe(01\bar{1}) + \alpha\text{-}Al_2O_3(0001)/\alpha\text{-}Fe(111) + nH\right] \quad (1)$$

where $n$ is the quantity of adsorbed H atoms at the $\alpha$-$Al_2O_3$(0001)/$\alpha$-Fe(111) interface. The terms of $\left[Fe(01\bar{1}) + \alpha\text{-}Al_2O_3(0001)/\alpha\text{-}Fe(111)\right]$ and $\left[Fe(01\bar{1}) + \alpha\text{-}Al_2O_3(0001)/\alpha\text{-}Fe(111)\right.$ $\left.+nH\right]$ are abbreviated as Fe(surface) and Fe(surface $+ nH_{ads}$), respectively. The change in free energy for the adsorption of $n$ H atoms at the interface under pressure $p$ and temperature $T$ is given by $\Delta G\left(p, T, \frac{1}{2}nH_2\right)$:

$$\Delta G\left(p, T, \frac{1}{2}nH_2\right) = G[Fe(surface + nH_{ads})] - G[Fe(surface)] - \frac{1}{2}nG[H_2] \quad (2)$$

where $G[Fe(surface + nH_{ads})]$ is the free energy of the $\left[Fe(01\bar{1}) + \alpha\text{-}Al_2O_3(0001)/\alpha\text{-}Fe(111)\right]$ plane with $n$ mol of adsorbed H atoms, $G[Fe(surface)]$ is the free energy of the clean Fe plane, and $G[H_2]$ is the free energy of gaseous $H_2$. The motional contribution of solid phases, e.g., the H-adsorbed Fe plane and the clean Fe plane, is negligible [31]. Thus, the free energy of the solid phases is replaced with the DFT-calculated energy:

$$\Delta G\left(p, T, \frac{1}{2}nH_2\right) = E[Fe(surface + nH_{ads})] - E[Fe(surface)] - \frac{1}{2}nG[H_2] \quad (3)$$

The free energy of $H_2$ molecules, $G[H_2]$, in Equation (3) is represented by the summation of the DFT-calculated energy and contributions of temperature and pressure [30,31]:

$$G[H_2] = E[H_2] + \widetilde{\mu}_{H_2} + RT \ln\frac{p_{H_2}}{p_0} \quad (4)$$

where $E[H_2]$ is the total energy of the $H_2$ molecules, the term $\widetilde{\mu}_{H_2}$ is the motional correction contributed by temperature, the term $RT\ln\frac{p_{H_2}}{p_0}$ is the motional correction contributed by gaseous pressure, $R$ is the ideal gas constant, $T$ is the temperature, $p_{H_2}$ is the partial pressure of $H_2$, and $p_0$ is the standard pressure. The temperature contribution can be derived as follows [61]:

$$\widetilde{\mu}_{H_2} = -k_B T N_A \ln Z \quad (5)$$

where $k_B$ is the Boltzmann constant, $N_A$ is Avogadro's constant, and $Z$ is the partition function, which is divided into three types of motion, i.e., 3D translational ($Z_{trans}^{3D}$), vibrational ($Z_{vib}$), and rotational ($Z_{rot}$). As $H_2$ is an ideal diatomic molecule, it can be treated as a harmonic oscillator. Thus, the three motions can be written as follows [62]:

$$Z_{trans}^{3D} = \left(\frac{2\pi m_{H_2}k_B T}{h^2}\right)^{\frac{3}{2}}\left(\frac{k_B T}{p_{H_2}}\right) \quad (6)$$

$$Z_{vib} = \frac{e^{-hv_{H_2}/2k_B T}}{1 - e^{-hv_{H_2}/k_B T}} \quad (7)$$

$$Z_{\text{rot}} = \frac{8\pi^2 I_{\text{H}_2} k_{\text{B}} T}{\sigma h^2} \tag{8}$$

where $m_{\text{H}_2}$ is the mass of H$_2$ molecules, $h$ is the Planck constant, $v_{\text{H}_2}$ is the stretch frequency of H–H bonds, $I_{\text{H}_2}$ is the moment of inertia for H$_2$, and $\sigma$ is a symmetrical factor (2 for diatomic molecules). Combining Equations (6)–(8) with Equation (5), the temperature correction term for H$_2$ molecules can be rewritten as follows:

$$\widetilde{\mu}_{\text{H}_2} = -k_{\text{B}} T N_A \ln\left( \left( \frac{2\pi m_{\text{H}_2} k_{\text{B}} T}{h^2} \right)^{\frac{3}{2}} \times \frac{k_{\text{B}} T}{p_{\text{H}_2}} \right) - k_{\text{B}} T N_A \ln\left( \frac{e^{-hv_{\text{H}_2}/2k_{\text{B}}T}}{1 - e^{-hv_{\text{H}_2}/k_{\text{B}}T}} \right) - k_{\text{B}} T N_A \ln\left( \frac{8\pi^2 I_{\text{H}_2} k_{\text{B}} T}{h^2} \right) \tag{9}$$

Combining Equations (3), (4) and (9), the change in free energy for the dissociative adsorption of $\frac{1}{2}n$ mol of H$_2$ at the $\alpha$-Al$_2$O$_3$(0001)/$\alpha$-Fe(111) interface on the Fe$(01\bar{1})$ plane can be obtained as follows:

$$
\begin{aligned}
\Delta G\left( p, T, \tfrac{1}{2}n\text{H}_2 \right) &= E[\text{Fe(surface)} + n\text{H}_{\text{ads}}] - E[\text{Fe(surface)}] \\
&\quad - \tfrac{1}{2}n \left( E[\text{H}_2] - k_{\text{B}} T N_A \ln\left( \left( \frac{2\pi m_{\text{H}_2} k_{\text{B}} T}{h^2} \right)^{\frac{3}{2}} \times \frac{k_{\text{B}} T}{p_{\text{H}_2}} \right) - k_{\text{B}} T N_A \ln\left( \frac{e^{-hv_{\text{H}_2}/2k_{\text{B}}T}}{1-e^{-hv_{\text{H}_2}/k_{\text{B}}T}} \right) \right. \\
&\quad \left. - k_{\text{B}} T N_A \ln\left( \frac{8\pi^2 I_{\text{H}_2} k_{\text{B}} T}{\sigma h^2} \right) + RT \ln \frac{p_{\text{H}_2}}{p_0} \right)
\end{aligned}
\tag{10}
$$

### 2.3. Changes in Free Energy for the Dissociative Adsorption of Gaseous O$_2$ and CH$_4$ Molecules

The effects of two types of gaseous components, O$_2$ and CH$_4$, within the transported fluid on the dissociative adsorption of hydrogen at the Al$_2$O$_3$ inclusion/Fe interface were investigated in this work. Thermodynamics analysis was performed to determine the feasibility of their associative adsorption at the interface. For dissociative adsorption of O$_2$ molecules [33]

$$\frac{1}{2}q\text{O}_2\,(\text{gas}) + \text{Fe(surface)} \rightarrow \text{Fe(surface} + q\text{O}_{\text{ads}}) \tag{11}$$

The change in free energy caused by the dissociative adsorption of O$_2$ at the $\alpha$-Al$_2$O$_3$(0001)/$\alpha$-Fe(111) interface on the Fe$(01\bar{1})$ plane, under given temperature and pressure, is as follows:

$$
\begin{aligned}
\Delta G\left( p, T, \tfrac{1}{2}q\text{O}_2 \right) &= E[\text{Fe(surface} + q\text{O}_{\text{ads}})] - E[\text{Fe(surface)}] \\
&\quad - \tfrac{1}{2}q \left( E[\text{O}_2] - k_{\text{B}} T N_A \ln\left( \left( \frac{2\pi m_{\text{O}_2} k_{\text{B}} T}{h^2} \right)^{\frac{3}{2}} \times \frac{k_{\text{B}} T}{p_{\text{O}_2}} \right) - k_{\text{B}} T N_A \ln\left( \frac{e^{-hv_{\text{O}_2}/2k_{\text{B}}T}}{1-e^{-hv_{\text{O}_2}/k_{\text{B}}T}} \right) \right. \\
&\quad \left. - k_{\text{B}} T N_A \ln\left( \frac{8\pi^2 I_{\text{O}_2} k_{\text{B}} T}{\sigma h^2} \right) + RT \ln \frac{p_{\text{O}_2}}{p_0} \right)
\end{aligned}
\tag{12}
$$

where $E_{\text{O}_2}$ is the DFT-calculated energy of O$_2$ molecules, $m_{\text{O}_2}$ is the mass of O$_2$ molecules, $p_{\text{O}_2}$ is the partial pressure of gaseous O$_2$, and $v_{\text{O}_2}$ is the stretch frequency of O–O bonds.

The dissociative adsorption of CH$_4$ is thermodynamically feasible at specific sites such as the HAGB and dislocations on the steel surface [18,19]. The dissociative adsorption of CH$_4$ is written as follows:

$$m\text{CH}_4\,(\text{gas}) + \text{Fe(surface)} \rightarrow \text{Fe(surface} + m\text{H}_{\text{ads}} + m\text{CH}_{3\text{ads}}) \tag{13}$$

After simplification of the free energy of solid phases into the DFT-calculated energy, the change in free energy of the CH$_4$ dissociative adsorption is as follows:

$$\Delta G(p, T, m\text{CH}_4) = E[\text{Fe(surface} + m\text{H}_{\text{ads}} + m\text{CH}_{3\text{ads}})] - E[\text{Fe(surface)}] - mG[\text{CH}_4] \tag{14}$$

As CH$_4$ is a polyatomic molecule, the calculation of its free energy should be modified by application of the partition function. The translational, vibrational, and rotational contributions of CH$_4$ are referred to in [62]. The change in free energy caused by the

dissociative adsorption of $CH_4$ at the $\alpha$-$Al_2O_3(0001)/\alpha$-$Fe(111)$ interface on the $Fe(01\bar{1})$ plane can be written as follows:

$$
\begin{aligned}
\Delta G(p,\ T,\ mCH_4) \quad &= E[\text{Fe(surface} + mH_{ads} + mCH_{3ads})] - E[\text{Fe(surface)}] \\
&-m\left( E[CH_4] - k_B T N_A \ln\left( \left(\frac{2\pi m_{CH_4} k_B T}{h^2}\right)^{\frac{3}{2}} \times \frac{k_B T}{p_{CH_4}} \right) - k_B T N_A \ln\left( \prod_{j=1}^{\alpha} \frac{e^{-\frac{\Theta_{vj}}{2T}}}{\left(1 - e^{-\frac{\Theta_{vj}}{T}}\right)} \right) \right. \\
&\left. -k_B T N_A \ln\left( \frac{\pi^{\frac{1}{2}}}{\sigma_{CH_4}} \left(\frac{T^3}{\Theta_A \Theta_B \Theta_C}\right)^{\frac{1}{2}} \right) + RT \ln\frac{p_{CH_4}}{p_0} \right)
\end{aligned}
\tag{15}
$$

where $\Theta_{vj}$ is the characteristic vibrational temperature, and $\Theta_A$, $\Theta_B$, and $\Theta_C$ are characteristic rotational temperatures.

### 2.4. Numerical Solution

The changes in free energy of the dissociative adsorption of $H_2$, $O_2$, and $CH_4$ molecules can be calculated by Equations (10), (12) and (15), respectively. The unknown terms in these equations include the DFT-calculated energies of $H_2$, $O_2$, and $CH_4$ molecules, of the H-adsorbed $Al_2O_3$/Fe interface, and of the bare $Al_2O_3$/Fe interface. The DMol$^3$ module in BOVIA Materials Studio 8.0 was applied to conduct spin-unrestricted DFT calculations to obtain the unknowns [63]. The generalized gradient approximation (GGA) and Perdew–Burke–Ernzerhof (PBE) functionals were used for correlation, as they can balance computational cost and modeling accuracy [64,65]. To expand the valence electron function into a set of numerical atomic orbitals, double numerical polarization (DNP) functions with a real space cutoff of 4.4 Å were applied. The convergence tolerance of the self-consistent field (SCF) was $1 \times 10^{-5}$ Ha/atom, and the energy convergence criterion was set to be $2 \times 10^{-5}$ Ha/atom. The maximum force and the displacement of atoms for every step during geometry optimization were set to 0.004 Ha/Å and 0.005 Å, respectively. The Brillouin-zone integrations were performed using $2 \times 4 \times 1$ and $3 \times 3 \times 1$ special Monkhorst–Pack k-point grids to sample the reciprocal space for the $\alpha$-$Al_2O_3(0001)/\alpha$-$Fe(111)$ interface on the $Fe(01\bar{1})$ plane and the $\alpha$-$Al_2O_3(0001)/\alpha$-$Fe(111)$ interface in bulk Fe, respectively. Furthermore, a sufficient $Z$-axis was applied to eliminate the influence of periodic boundary conditions on mutual H interactions. The supercell sizes for the plane model and the interface model were X = 28.6430 Å, Y = 14.1597 Å, Z = 22.1310 Å and X = Y = 8.1751 Å, Z = 25.2278 Å, respectively.

## 3. Results and Discussion

### 3.1. Configurations of Dissociative Adsorption of Hydrogen at the $\alpha$-$Al_2O_3(0001)/\alpha$-$Fe(111)$ Interface on the $Fe(01\bar{1})$ Plane

The $\alpha$-$Al_2O_3(0001)/\alpha$-$Fe(111)$ interface on the $Fe(01\bar{1})$ plane and the sites for stable adsorption of hydrogen are shown in Figure 3, where five hydrogen adsorption sites were labeled after consideration of the periodic microstructure and elimination of unstable configurations. There are three sites on the Fe side and two sites on the $Al_2O_3$ side.

The H adsorption energy, $E_{ads}^H$, can be calculated as follows [23,31]:

$$
E_{ads}^H = E[\text{Fe(surface} + H_{ads})] - E[\text{Fe(surface)}] - \frac{1}{2}E[H_2]
\tag{16}
$$

The calculated $E_{ads}^H$ at the $\alpha$-$Al_2O_3(0001)/\alpha$-$Fe(111)$ interface on the $Fe(01\bar{1})$ plane is shown in Figure 4. For comparison, the $E_{ads}^H$ on the Fe(100) plane, HAGB, and dislocation is also labeled [18,19]. It can be seen that the $E_{ads}^H$ varies between 0.14 eV and $-0.78$ eV. When H atoms adsorb on the Fe side of the interface, the $E_{ads}^H$ is more negative (i.e., $-0.60$ eV, $-0.71$ eV, and $-0.78$ eV at Sites #1, #2, and #3, respectively) than for H adsorption on the $Al_2O_3$ side (i.e., $-0.38$ eV and 0.14 eV at Sites #4 and #5, respectively). The results indicate that the stability of the H atom adsorption at the $\alpha$-$Al_2O_3(0001)/\alpha$-$Fe(111)$ interface is

not uniform. The Fe side is more stable for the dissociative adsorption of hydrogen than the Al$_2$O$_3$ side. Furthermore, the $E_{ads}^H$ on the Fe(100) plane, HAGB, and dislocation are $-0.44$ eV, $-0.66$ eV, and $-0.73$ eV, respectively. The sites at the interface with negative $E_{ads}^H$, especially those on the Fe side, are preferential sites for the dissociative adsorption of hydrogen, as compared to the Fe(100) plane. The stability of the H adsorption on the Fe side is like the HAGB and dislocation defects. It is thus expected that, in high-pressure gaseous environments, dissociative adsorption of hydrogen tends to occur at the Al$_2$O$_3$/Fe interface, especially on the Fe side. This implies that the inclusion/steel matrix interface is a potential site to accumulate H atoms, initiating cracks under given stress conditions.

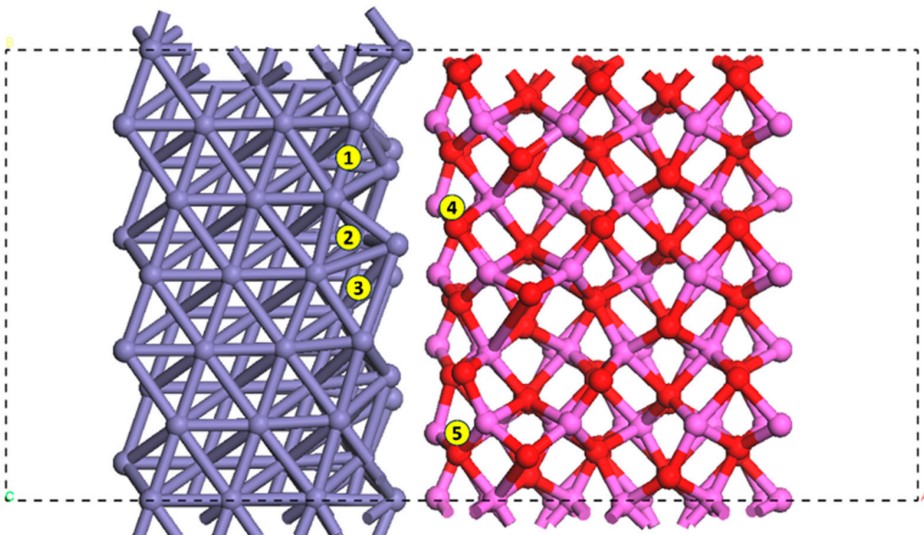

**Figure 3.** Sites for stable hydrogen adsorption at the $\alpha$-Al$_2$O$_3$(0001)/$\alpha$-Fe(111) interface on the Fe$(01\bar{1})$ plane. The box with dashed lines is the supercell used in this study. Purple: Fe atoms. Pink: Al atoms. Red: O atoms. Site #1: long bridge. Sites #2 and #3: threefold. Site #4: O site. Site #5: Al site.

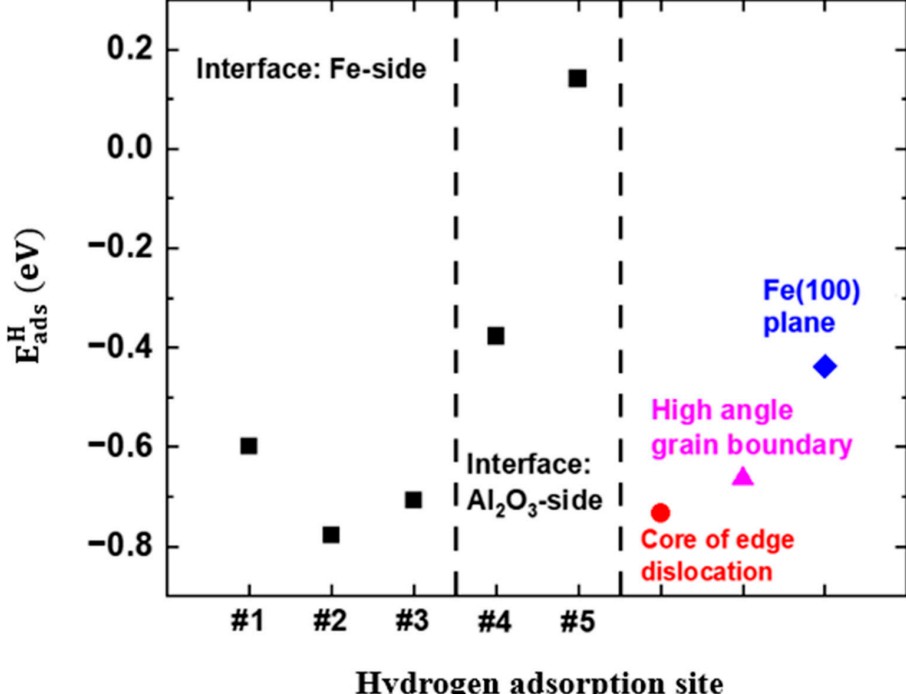

**Figure 4.** The H adsorption energies at various sites at the $\alpha$-Al$_2$O$_3$(0001)/$\alpha$-Fe(111) interface on the Fe$(01\bar{1})$ plane.

### 3.2. Thermodynamics of Dissociative Adsorption of Hydrogen at the α-Al₂O₃(0001)/α-Fe(111) Interface on the Fe(01̄1̄) Plane

The DFT-calculated energies of the clean Fe plane, isolated H₂ molecules, and hydrogen-adsorbed Fe plane are used in Equation (10). The free-energy profiles are plotted with temperature as the *X*-axis and pressure as the *Y*-axis. The changes in free energy of the dissociative adsorption of hydrogen at the five sites around the α-Al₂O₃(0001)/α-Fe(111) interface on the Fe(01̄1̄) plane, along with their results, are shown in Figure 5. The operating pressure of hydrogen pipelines ranges from 500 to 1200 psi (i.e., 3.45–8.27 MPa) [66], and the temperature is between 15 and 50 °C (i.e., 288.15–323.15 K) [67]. Thus, the pressure and temperature used for thermodynamics calculations in this work were selected to be 3–9 MPa and 283.15–323.15 K, respectively. It can be seen that the changes in free energy at Sites #1–#4 are between −0.036 eV and −0.513 eV, indicating that the dissociative adsorption of hydrogen at these sites is thermodynamically feasible. However, the changes in free energy at Site #5 are between 0.4058 eV and 0.4824 eV. The positive values indicate that the dissociative adsorption of hydrogen at Site #5 is thermodynamically infeasible unless additional energy is provided [68]. Moreover, the change in free energy becomes more negative as the partial pressure of H₂ increases and temperature decreases. These results suggest favorable conditions for the occurrence of dissociative adsorption of hydrogen. Elevated pressure and decreased temperature can stabilize the configurations of the dissociative adsorption of hydrogen via enhancing the contribution from isolated H₂ motion, resulting in an additional drop in free energy. Of the sites with negative free-energy changes, Site #2 exhibits the most negative free-energy change, followed by Sites #3, #1, and #4. Therefore, the dissociative adsorption of H₂ molecules preferentially occurs on the Fe side of the interface.

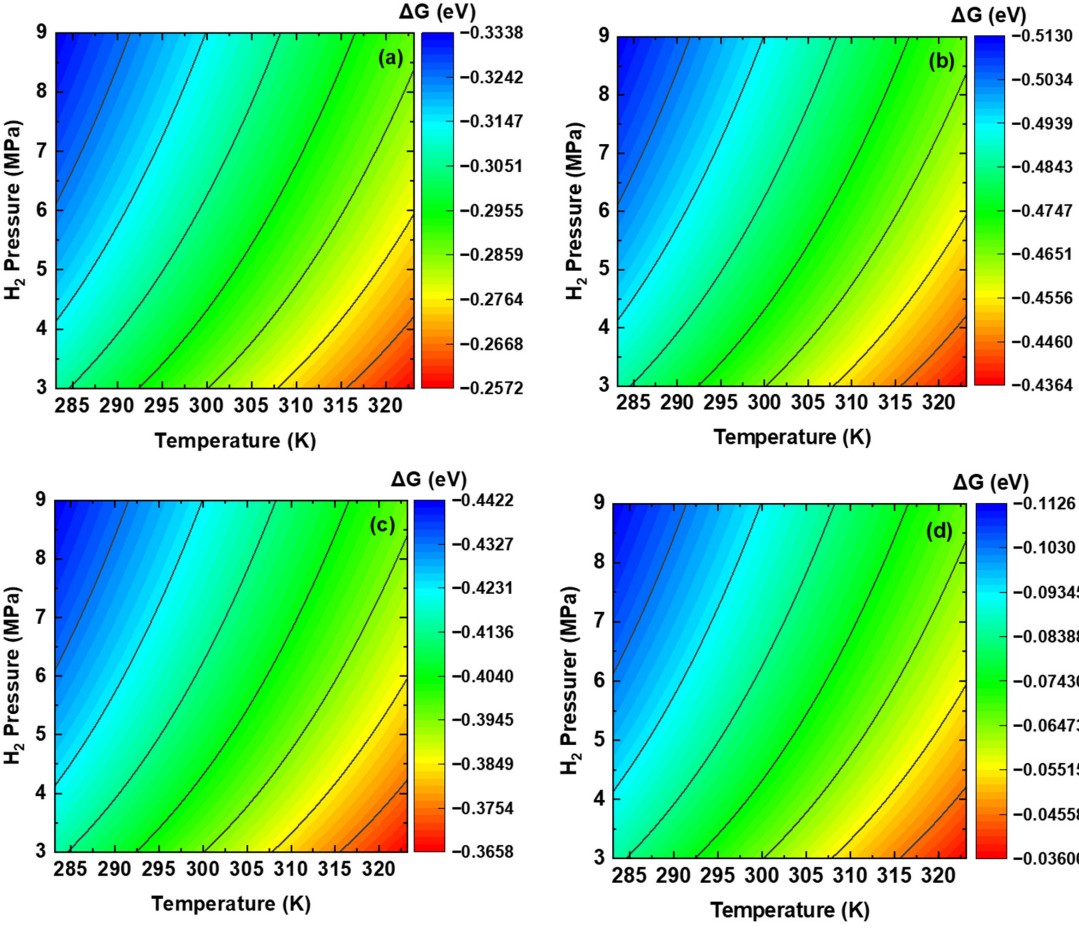

**Figure 5.** *Cont.*

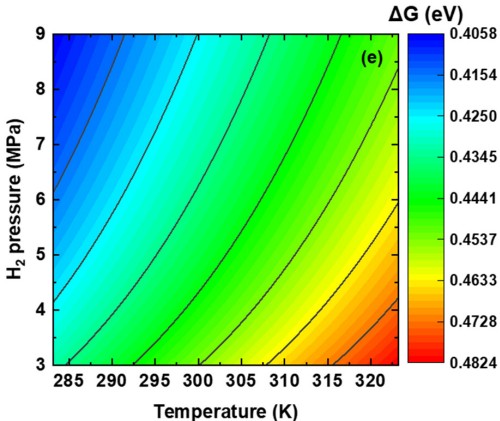

**Figure 5.** Changes in free energy of the dissociative adsorption of hydrogen at the five sites of the $\alpha$-Al$_2$O$_3$(0001)/$\alpha$-Fe(111) interface on the Fe$\left(01\bar{1}\right)$ plane under pipeline operating conditions: (**a**) Site #1, (**b**) Site #2, (**c**) Site #3, (**d**) Site #4, and (**e**) Site #5.

*3.3. Dissociative Adsorption of H$_2$ Molecules at the $\alpha$-Al$_2$O$_3$(0001)/$\alpha$-Fe(111) Interface*

The dissociative adsorption of H$_2$ molecules, along with the subsequent bonding between the generated H atoms and Fe atoms on crystalline planes, as well as on grain boundaries and dislocations on the surface of steels, occurs through orbital hybridization facilitated by electron shifting [13,18,19,25,30,33]. The electron densities of the interface in the presence and absence of adsorbed H atoms were determined by DFT modeling based on orbital hybridization. The electron density difference, namely, the shift in charges during the dissociative adsorption of hydrogen, at various sites at the $\alpha$-Al$_2$O$_3$(0001)/$\alpha$-Fe(111) interface is shown in Figure 6, where blue and yellow indicate electron accumulation and electron consumption, respectively. On the Fe side (i.e., Sites #1, #2, and #3), electrons shift from the Fe atoms to the H atoms, leading to electron accumulation at the H atoms, while the Fe atoms experience electron depletion. The charged H atoms repel each other and cause cleavage of the H–H bonds. On the Al$_2$O$_3$ side (i.e., Sites #4 and #5), the electron shift occurs between the H atoms and the O and Al atoms at Sites #4 and #5, respectively. At Site #4, the electron accumulation and consumption are observed at the H and O atoms, respectively. The H–Al bond at Site #5 features electron accumulation and consumption at the H atoms and Al atoms, respectively. Moreover, the O atoms adjacent to the top Al atoms display charge accumulation due to their high electronegativity [33]. Thus, the O atoms can compete with H atoms to capture electrons from the Al atoms, resulting in weak H adsorption. This can explain the less negative $E_{\text{ads}}^{\text{H}}$ and the less stable H adsorption configuration at Site #5.

The partial density of states (PDOS) of hydrogen adsorption at various sites of the $\alpha$-Al$_2$O$_3$(0001)/$\alpha$-Fe(111) interface was analyzed, and the results are shown in Figure 7. The orbitals from Al and H atoms in Figure 7c,d are plotted on the right *Y*-axis to avoid magnitude differences. For hydrogen adsorption on the Fe side, hybridization is formed among the Fe *s* orbital, Fe *d* orbital, and H *s* orbital. The H *s* orbital at Site #1 primarily overlaps with the Fe *s* orbital and the Fe *d* orbital at $-5.08$ eV. At Site #2 on the Fe side, the primary overlapping peaks among the H *s* orbital, Fe *s* orbital, and Fe *d* orbital are observed at $-6.42$ eV, suggesting that the hybridization is formed at a more negative potential energy. Moreover, a secondary hybridization peak is found at $-5.51$ eV. Thus, Site #2 displays a more negative $E_{\text{ads}}^{\text{H}}$ and a more stable hydrogen adsorption configuration than Site #1. For the hydrogen adsorption on the Al$_2$O$_3$ side, one hybridization peak between Al and O atoms is located at a very deep energy level of approximately $-20$ eV, showing good consistency with previous work [69]. This is caused by the high electronegativity of O atoms and the high tendency of electron loss at Al atoms. Thus, the Al$_2$O$_3$ inclusion exhibits strong and stable Al–O bonds, reducing the dissociative adsorption of hydrogen. As a result, the $E_{\text{ads}}^{\text{H}}$ at the Al$_2$O$_3$ is less negative, at $-0.15$ eV. The H atoms adsorbed at Site #4

are hybridized with the O *s* orbital and Al *s* orbital at $-21.24$ eV. Another overlapping peak among the H *s* orbital, O *p* orbital, and Al *s* and *p* orbitals is observed at $-9.40$ eV, and a small Al–H hybridization peak is found at 2.04 eV. The hybridization peak for H adsorption at Site #5 overlaps with the Al *s* and *p* orbitals and the O *p* orbital at a less negative energy of $-1.97$ eV, indicating an unstable adsorption configuration compared to the H adsorption at Site #4.

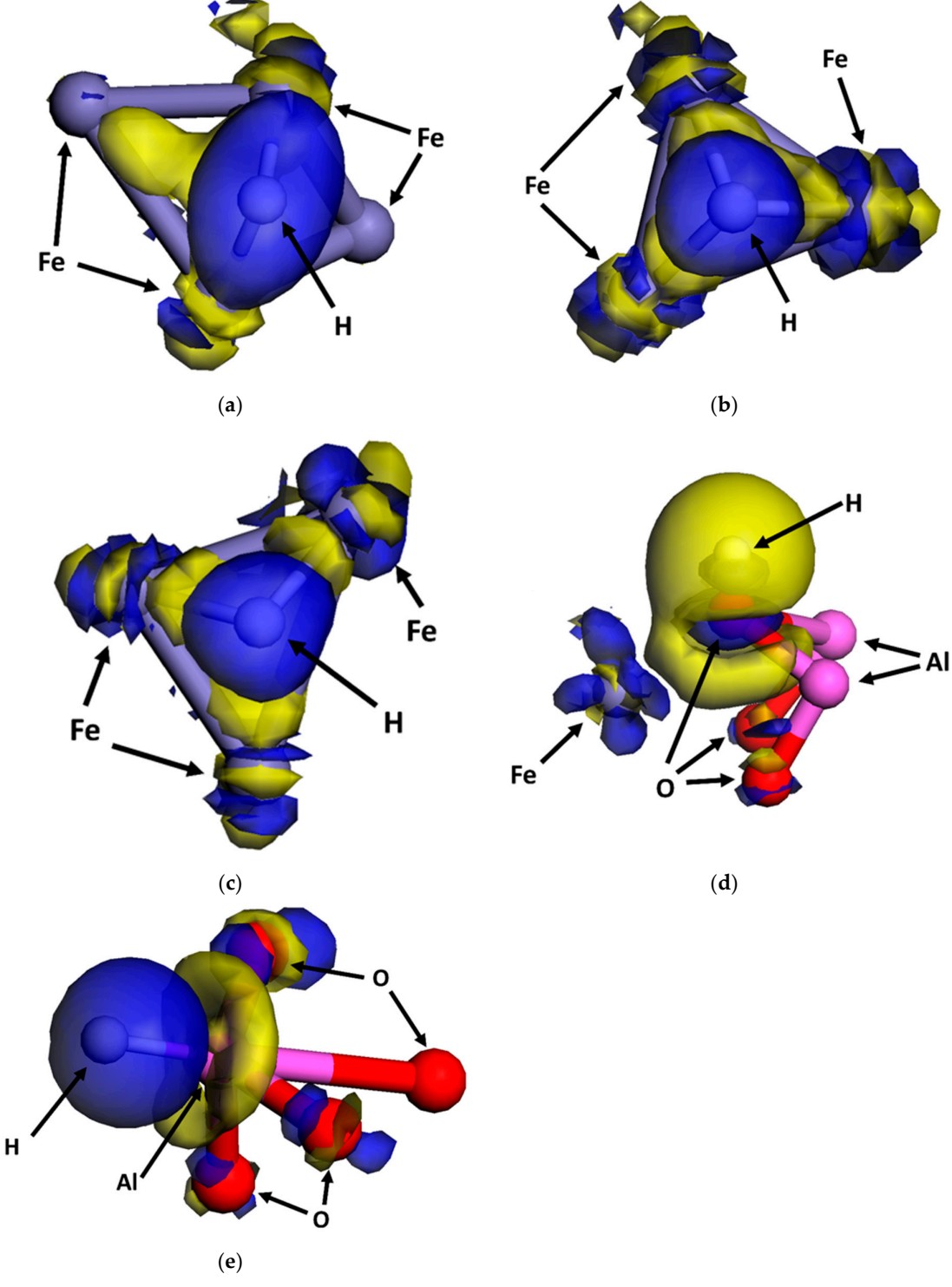

**Figure 6.** Electron density difference for hydrogen adsorption at various sites of the $\alpha$-Al$_2$O$_3$(0001)/$\alpha$-Fe(111) interface: (**a**) Site #1, (**b**) Site #2, (**c**) Site #3, (**d**) Site #4, and (**e**) Site #5. Blue: electron accumulation. Yellow: electron consumption.

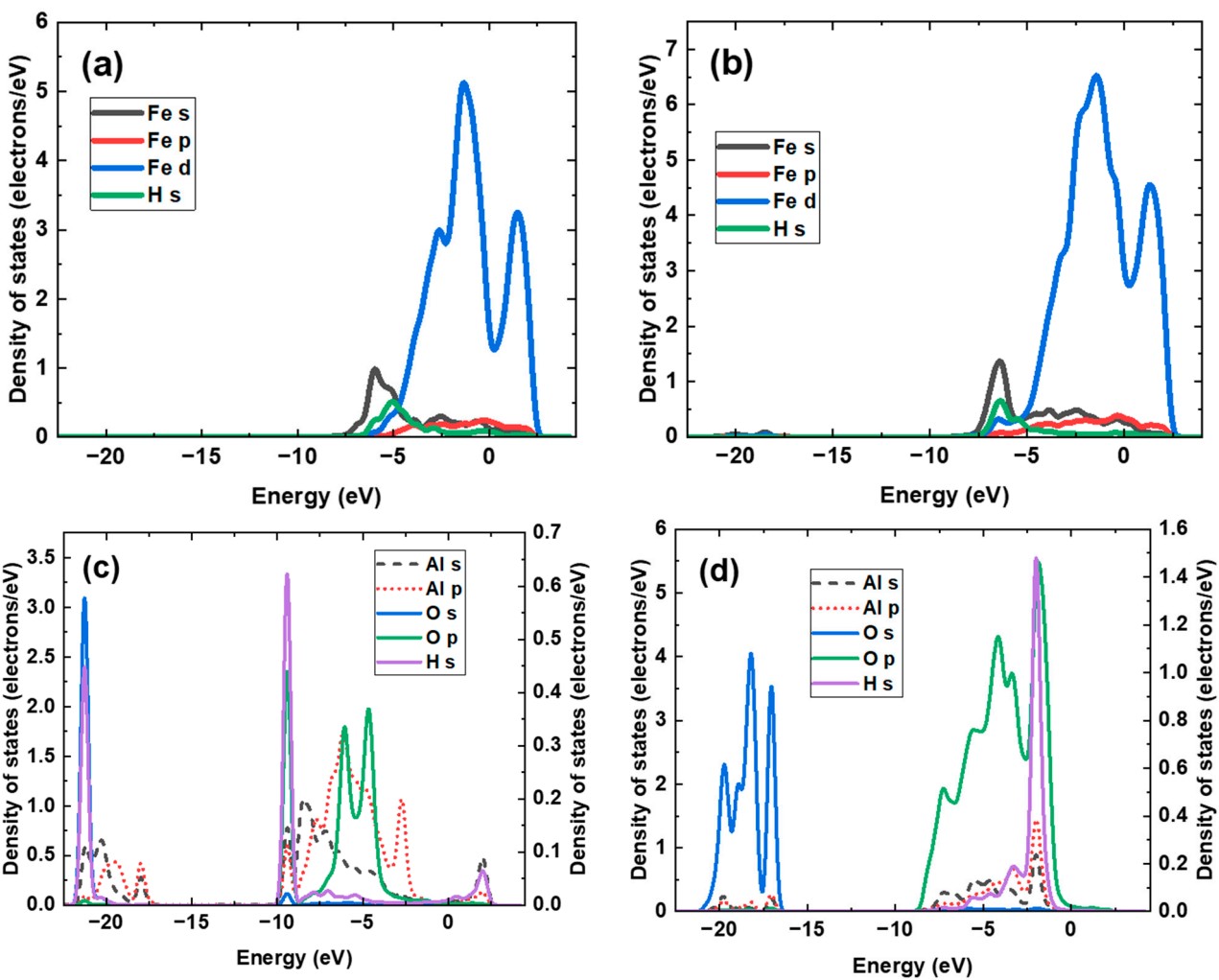

**Figure 7.** Partial density of states of dissociative adsorption of hydrogen at some typical sites of the α-Al₂O₃(0001)/α-Fe(111) interface: (**a**) Site #1, (**b**) Site #2, (**c**) Site #4, and (**d**) Site #5.

*3.4. Dissociative Adsorption of CH₄ and O₂ Molecules at the α-Al₂O₃(0001)/α-Fe(111) Interface*

To investigate the dissociative adsorption of $CH_4$ and $O_2$ gaseous molecules at the interface, the adsorbent is limited to being one non-repetitive unit: one O atom for $O_2$ adsorption, and one -CH₃ complex and one H atom for $CH_4$ adsorption, to save computational time. The configurations of the dissociative adsorption of $O_2$ and $CH_4$ molecules at the α-Al₂O₃(0001)/α-Fe(111) interface are shown in Figure 8. The adsorption energies of -CH₃ and O atoms are calculated as follows:

$$E_{ads}^{-CH_3} = E[Fe(surface + H_{ads} + CH_{3ads})] - E[Fe(surface)] - E[CH_4] \qquad (17)$$

$$E_{ads}^{O} = E[Fe(surface + O_{ads})] - E[Fe(surface)] - \frac{1}{2}E[O_2] \qquad (18)$$

The $E_{ads}^{-CH_3}$ and $E_{ads}^{O}$ were calculated to be 0.04 eV and −3.13 eV, respectively. Thus, the dissociative adsorption of $CH_4$ at the Al₂O₃/Fe interface is unstable. However, a stable adsorption configuration of O atoms can be established. The O adsorption energy is more negative than the H adsorption energy, indicating the preferential adsorption of $O_2$ over $H_2$ at the Al₂O₃/Fe interface.

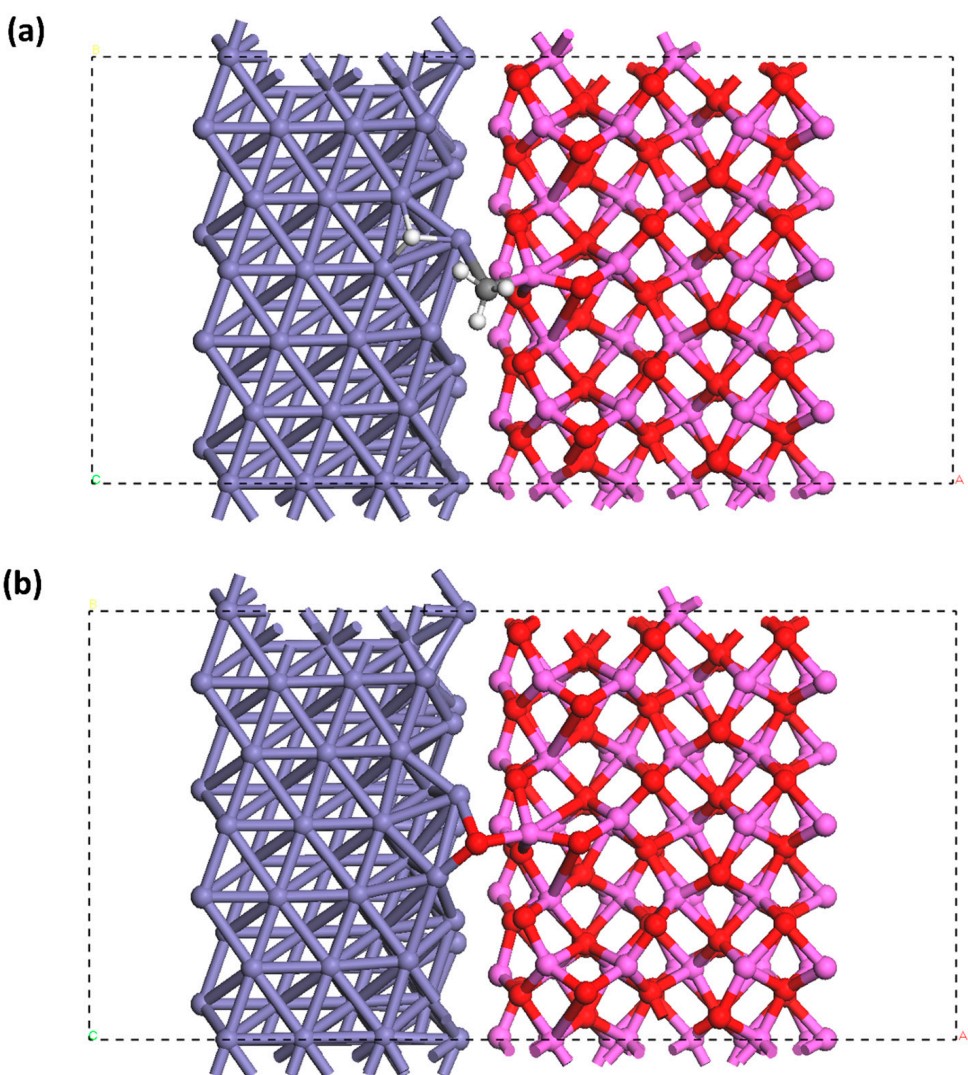

**Figure 8.** Configurations of the dissociative adsorption of (**a**) $CH_4$ and (**b**) $O_2$ at the $\alpha$-$Al_2O_3(0001)/$ $\alpha$-Fe(111) interface on the Fe$(01\bar{1})$ plane.

The changes in free energy for associative adsorption of $CH_4$ and $O_2$ molecules at the $Al_2O_3/Fe$ interface were calculated using the partition functions in Equations (12) and (15) under varied temperatures and pressures, and the results are shown in Figure 9. The temperature and pressure ranges were set in alignment with the operating conditions of hydrogen pipelines [70]. It can be seen that the dissociative adsorptions of both $CH_4$ and $O_2$ molecules at the $Al_2O_3/Fe$ interface display negative changes in free energy, indicating the thermodynamic feasibility of the process. Elevated partial pressure of hydrogen and reduced temperature favor the dissociative adsorption of $CH_4$ and $O_2$. The changes in free energy of both $CH_4$ and $O_2$ are more negative than that of $H_2$, suggesting the competitive preference of the dissociative adsorption of $CH_4$ and $O_2$ over $H_2$. Particularly, the most negative free-energy change of $-2.585$ eV~$-2.371$ eV was observed for the dissociative adsorption of $O_2$.

Electron transfer during bonding between Fe atoms and -$CH_3$ complexes and O atoms at the $Al_2O_3/Fe$ interface is determined by the electron density difference generated during orbital hybridization through DFT modeling, as shown in Figure 10. After dissociation, the $CH_4$ molecules generate H atoms and -$CH_3$ complexes. Electrons shift from the Fe atoms to the generated H atoms. The electron transfer at the -$CH_3$ complex is complicated. Electron depletion occurs at Fe atoms and Al atoms, and electron accumulation occurs at the C

atom in the -CH$_3$ complex. The dissociative adsorption of O$_2$ molecules leads to electron accumulation at the adsorbed O atoms and electron depletion at Al atoms and Fe atoms.

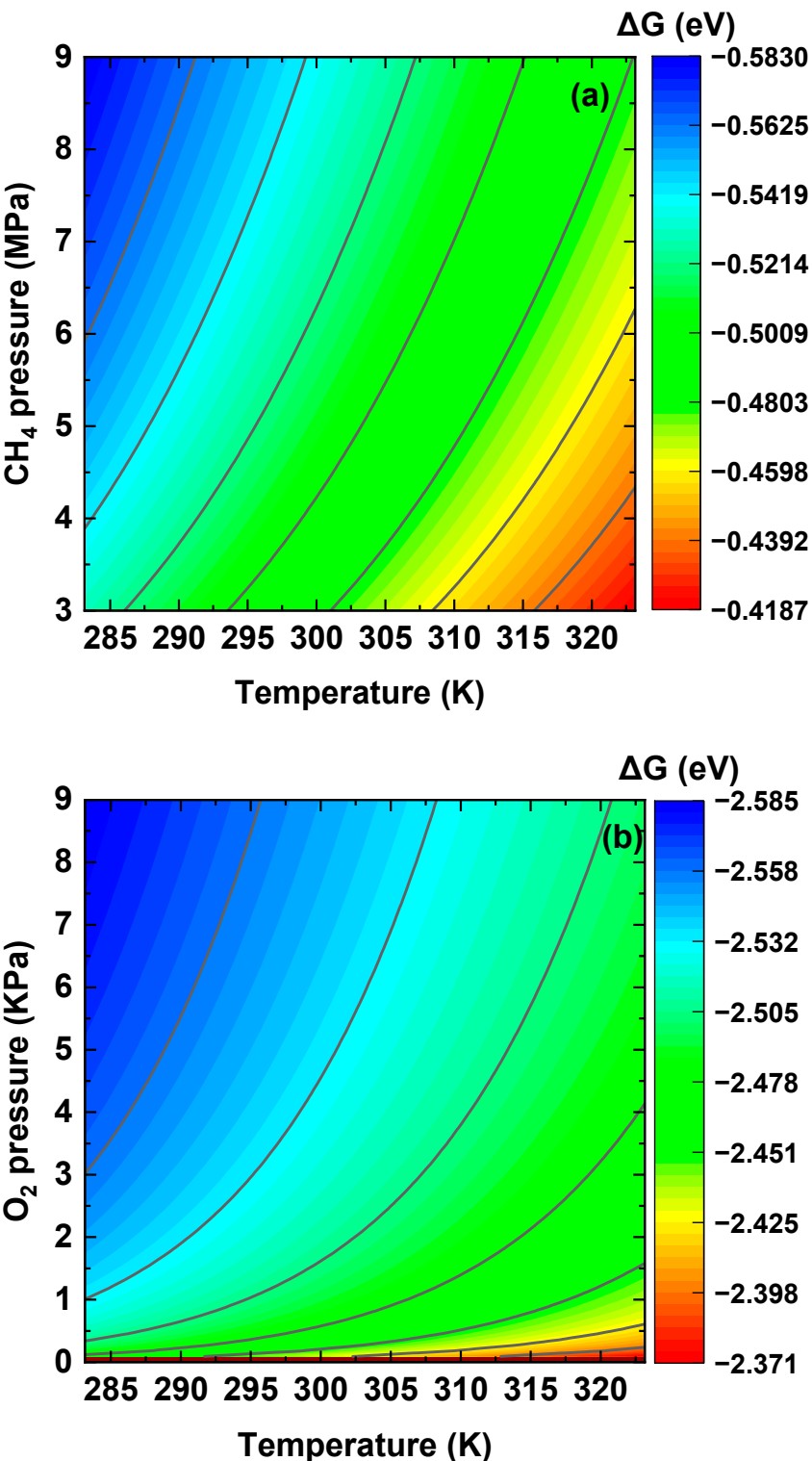

**Figure 9.** Changes in free energy for dissociative adsorption of (**a**) CH$_4$ and (**b**) O$_2$ at the $\alpha$-Al$_2$O$_3$(0001)/ $\alpha$-Fe(111) interface on the Fe(01$\bar{1}$) plane under pipeline operating conditions.

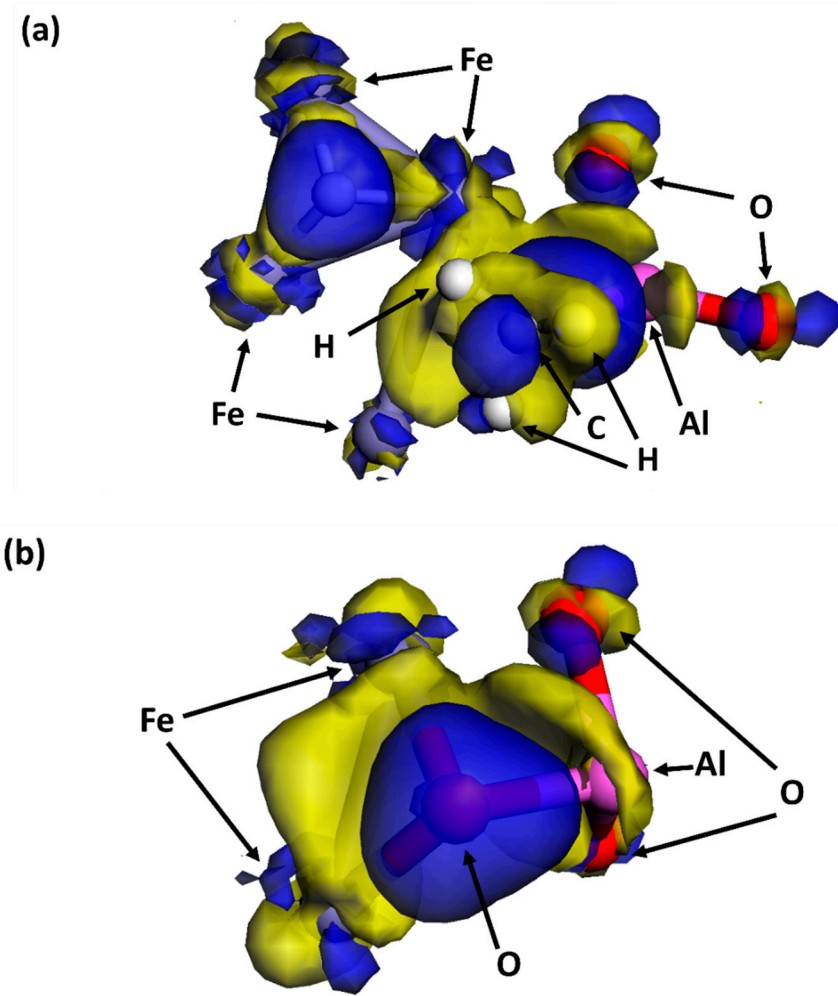

**Figure 10.** Electron density difference for dissociative adsorption of (**a**) CH$_4$ and (**b**) O$_2$ at the $\alpha$-Al$_2$O$_3$(0001)/$\alpha$-Fe(111) interface on the Fe(01$\bar{1}$) surface. Blue: electron accumulation. Yellow: electron consumption.

The PDOS of dissociative adsorption of CH$_4$ and O$_2$ molecules at the $\alpha$-Al$_2$O$_3$(0001)/$\alpha$-Fe(111) interface is calculated from the electron density distributions in Figure 10, and the results are shown in Figure 11, where Al orbitals are plotted to the right $Y$-axis to avoid magnitude differences with other orbitals. Both H and C atoms from the -CH$_3$ complex hybridize with Al atoms at $-12.43$ eV, where the overlapping peaks can be observed among the C $s$, H $s$, and Al $s$ orbitals. Another hybridization peak among C, H, and Al atoms is observed at $-5.40$ eV, where the hybridization is formed by C $p$, H $s$, and Al $p$ orbitals. In addition, the C $p$ orbital also hybridizes with Al $s$ and $p$ orbitals at $-2.17$ eV. Some hybridization peaks with a low intensity among Fe and C atoms are detected at $-12.43$ eV, $-5.40$ eV, and $-2.17$ eV, demonstrating that the strength of the Fe–C bond is weak. For the dissociative adsorption of O$_2$, the hybridization peak between the Al $p$ orbital and O $s$ orbital is observed at a deep energy level of $-18.14$ eV. Moreover, multiple overlapping peaks among Al and adsorbed O atoms are found between $-7.5$ eV and $-1.0$ eV. It should be noted that Fe atoms participate in hybridization with the adsorbed O atoms, as the Fe $s$ orbital overlaps with the O $s$ orbital at $-18.14$ eV. The Fe $d$ orbital overlaps with the O $p$ orbital at $-4.84$ eV. A negative O adsorption energy and a strong O adsorption configuration are achieved.

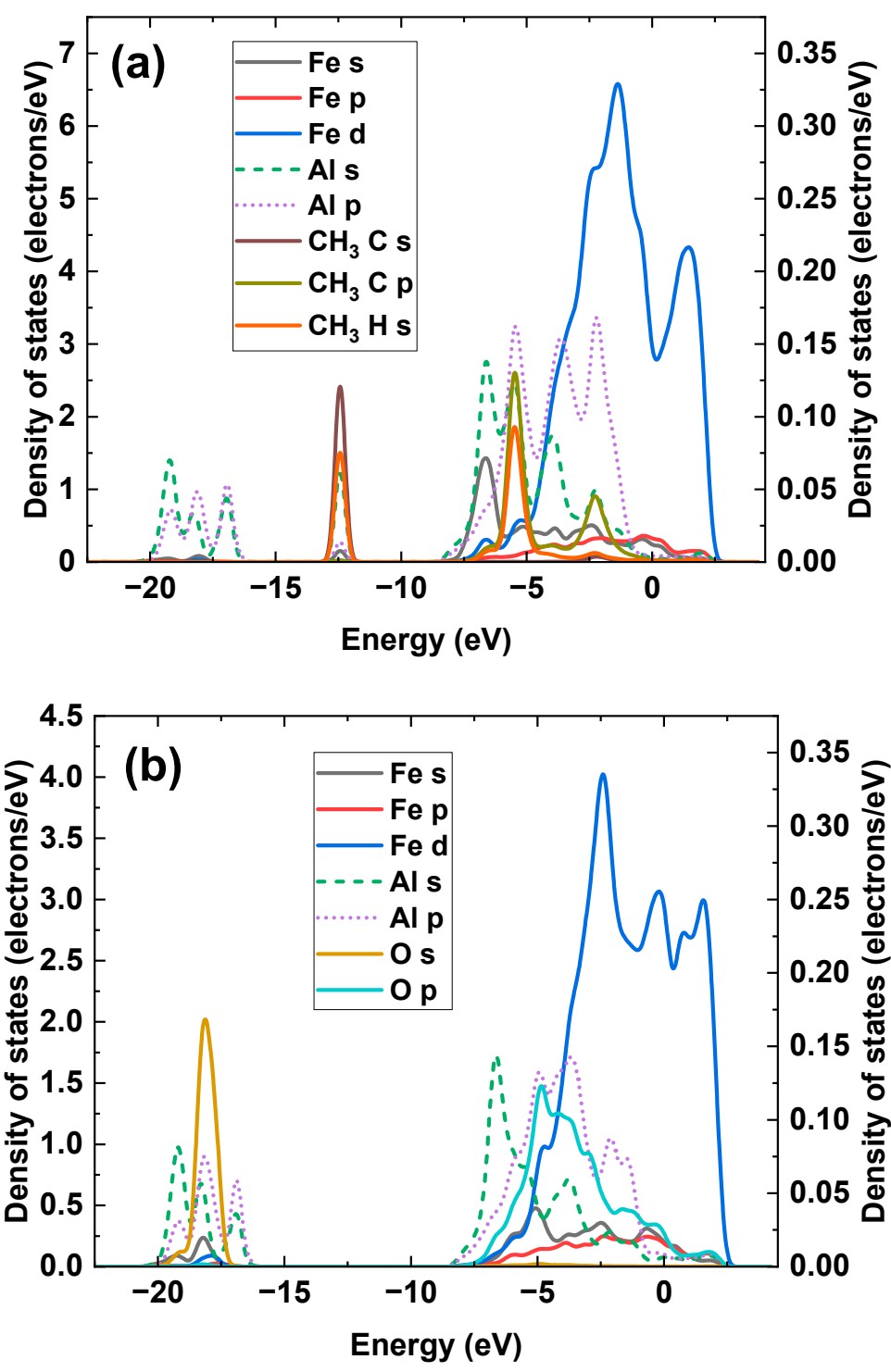

**Figure 11.** The PDOS of dissociative adsorption of (**a**) CH$_4$ and (**b**) O$_2$ at the $\alpha$-Al$_2$O$_3$(0001)/$\alpha$-Fe(111) interface.

### 3.5. Hydrogen Atom Accumulation at the $\alpha$-Al$_2$O$_3$(0001)/$\alpha$-Fe(111) Interface

Figure 12 shows the configurations of the H atom accumulation at the $\alpha$-Al$_2$O$_3$(0001)/$\alpha$-Fe(111) interfaces with different terminations. The trapped H atoms are found at the quasi-4-fold site. The H binding energy, $E_{\text{binding}}^{\text{H}}$, is calculated as the difference between the energy of the interface with trapped H atoms and the total energy of the interface without H atoms and the H atoms [71]. The $E_{\text{binding}}^{\text{H}}$ of the Al-terminated and O-terminated interfaces was calculated to be 0.33 eV and 0.18 eV, respectively, consistent with the H binding energy

of 0.24 eV found in a previous work [44]. The $E_{binding}^{H}$ at the Al$_2$O$_3$/Fe interface is much greater than that of the crystalline planes [13], indicating the strong H-trapping effect of the Al$_2$O$_3$ inclusion/Fe interface. The H trapping at the Al-terminated interface is stronger than at the O-terminated interface. This is caused by the lower adhesion work of the Al-terminated interface [46]. The PDOS of H trapping at the $\alpha$-Al$_2$O$_3$(0001)/$\alpha$-Fe(111) interface with different terminations was calculated from the electron density distributions with specific configurations, as shown in Figure 13. It can be seen that primary hybridization peaks at the Al-terminated and the O-terminated interfaces are at $-7.32$ eV and $-5.38$ eV, respectively. The hybridization for H atoms trapped at the Al-terminated interface is formed at a deeper energy level, indicating a stronger effect.

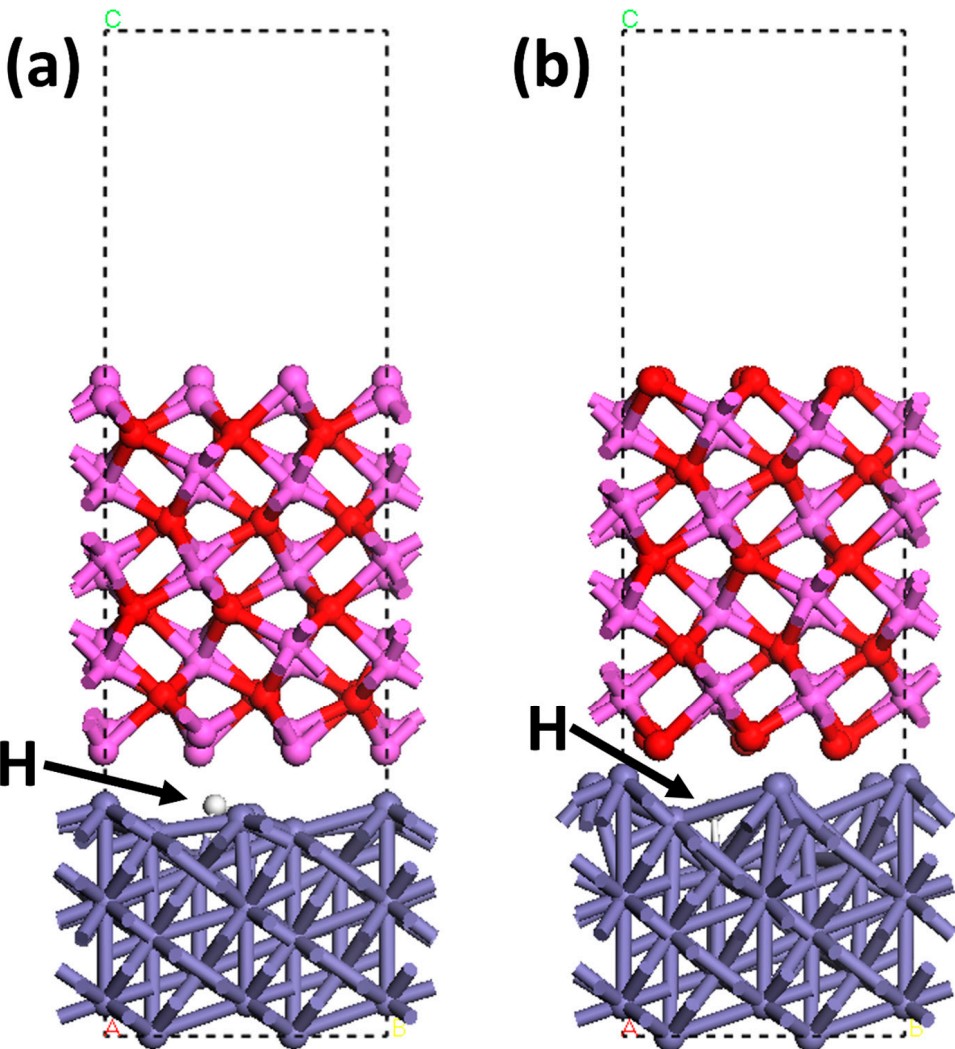

**Figure 12.** Configurations of the H atom trapping at the $\alpha$-Al$_2$O$_3$(0001)/$\alpha$-Fe(111) interfaces with different terminations: (**a**) Al-terminated; (**b**) O-terminated. Purple: Fe atom. Pink: Al atom. Red: O atom. White: H atom.

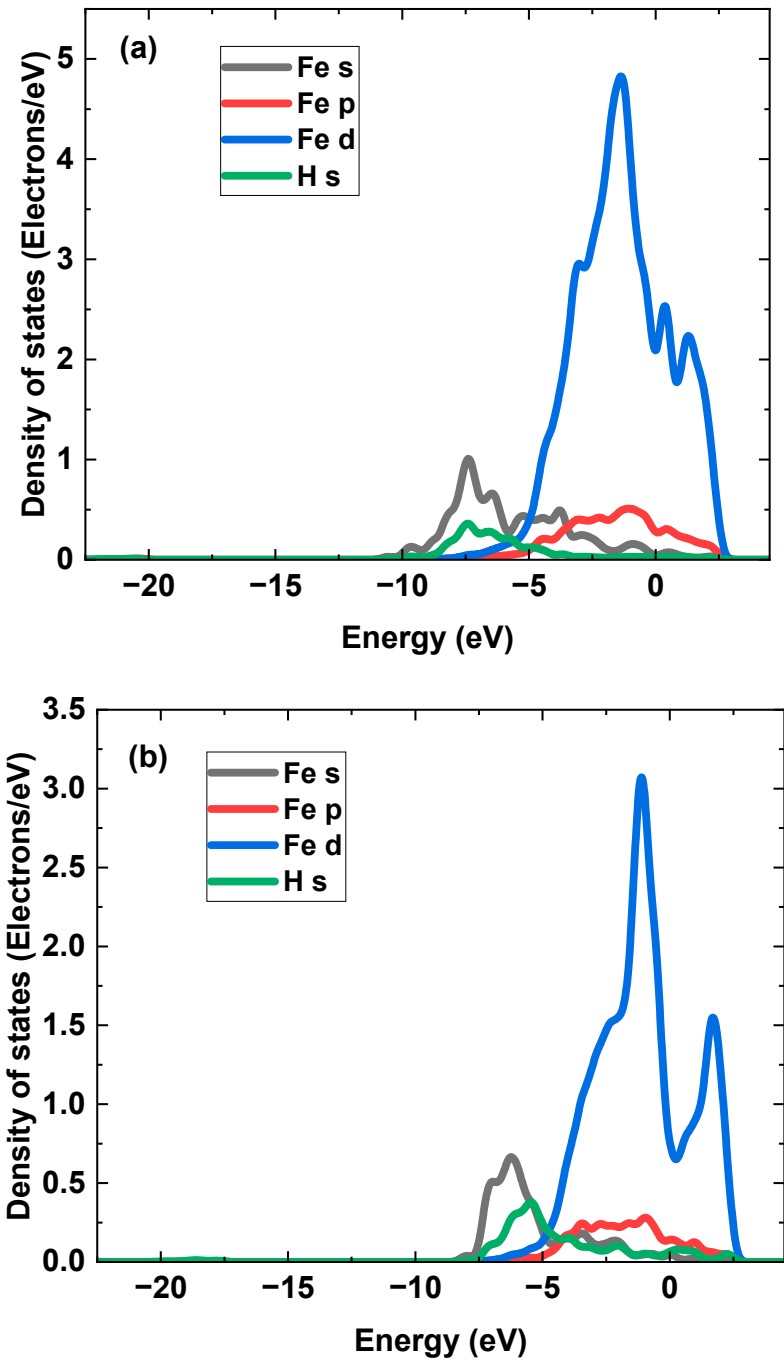

**Figure 13.** The PDOS of H atom trapping at the $\alpha$-Al$_2$O$_3$(0001)/$\alpha$-Fe(111) interfaces with (**a**) Al termination and (**b**) O termination.

## 4. Conclusions

This work explores the dissociative adsorption of hydrogen at the $\alpha$-Al$_2$O$_3$(0001)/$\alpha$-Fe(111) interface on the Fe$(01\bar{1})$ plane, determining the H$_2$ molecule adsorption, dissociation, and H atom adsorption configurations at various sites of the interface. The occurrence of dissociative adsorption of hydrogen at the Al$_2$O$_3$ inclusion/Fe interface is favored under conditions of elevated partial pressure of H$_2$ and low temperature. Under pipeline operating conditions, the thermodynamic feasibility of dissociative adsorption of hydrogen is observed for Fe and O atoms. However, such feasibility is not observed for Al atoms under these conditions.

On the Fe side of the interface, a more negative H adsorption energy was observed compared to the $Al_2O_3$ side. This result suggests that H atoms can form more stable adsorption configurations on the Fe side. It is expected that H atoms, generated through the dissociative adsorption of $H_2$ molecules, will exhibit a preference for accumulation on the Fe side of the $Al_2O_3$ inclusion/Fe interface. In contrast, the $Al_2O_3$ side of the interface exhibits similar or less negative H adsorption energy than that of the crystalline planes. Thus, it is unlikely that H atoms will be able to adsorb on the $Al_2O_3$ side of the interface. H atoms adsorbing onto Fe and Al atoms induce electron attraction from both Fe and Al. Conversely, H atoms adsorbing onto O atoms undergo a bidirectional charge shift. H–Fe bonds are established through the orbital hybridization mechanism. The robust O–Al bonding strength within the $Al_2O_3$ inclusion weakens the overall strength of H adsorption, rendering the adsorption on the $Al_2O_3$ side of the interface unstable.

The presence of the impurity gases of $O_2$ and $CH_4$ within the fluid impacts the dissociative adsorption of hydrogen at the $Al_2O_3$ inclusion/Fe interface. The dissociative adsorption of $O_2$ and $CH_4$ at the interface is thermodynamically feasible. The changes in free energy for the dissociative adsorption of $CH_4$ and $O_2$ are more negative compared to the dissociative adsorption of hydrogen. This suggests a greater tendency for the occurrence of dissociative adsorption of $O_2$ and $CH_4$ over $H_2$ due to their more favorable energetics. Particularly, the dissociative adsorption of $O_2$ is preferential over $CH_4$ due to the more negative change in free energy. Due to its more negative adsorption energy and the deeper energy level of the hybridization peaks, the formed O–Fe bond is inherently more stable than the $CH_3$–Fe bond. During bonding, electrons shift to the adsorbents of O atoms and -$CH_3$ complexes from Fe, Al, and O atoms at the interface. The Al-terminated interface exhibits a higher H binding energy compared to the O-terminated interface, indicating a preference for hydrogen accumulation at the Al-terminated interface.

**Author Contributions:** Conceptualization, F.C.; Methodology, Y.S.; Formal analysis, Y.S.; Investigation, Y.S.; Writing—review & editing, F.C.; Supervision, F.C.; Project administration, F.C.; Funding acquisition, F.C. All authors have read and agreed to the published version of the manuscript.

**Funding:** This research was supported by Alberta Innovates (Project no. 222300977). Productive discussions with Yine Ren of Dalian University of Technology are appreciated.

**Data Availability Statement:** Data are available upon request.

**Conflicts of Interest:** The authors declare no conflicts of interest.

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
