# Peer review of "Dissociative Adsorption of Hydrogen Molecules at Al2O3 Inclusions in Steels and Its Implications for Gaseous Hydrogen Embrittlement of Pipelines"

_cmd, doi:10.3390/cmd5020008_

Round 1

Reviewer 1 Report

Comments and Suggestions for Authors

The manuscript targets a very interesting topic that is the hydrogen embrittlement. I went through the text and I believe that contains interesting data. However, after reading it several times I believe that the manuscript can be published only after major changes or new resubmission. 

The introduction is quite ok and briefly describes the problematic of the hydrogen embrittlement which is an important topic as hydrogen may serve as a future source of clean energy. Also, the references section contains relevant data. However, some parts of the text are not clean, especially figures.

Line 135 and 154: Both figures (Fig. 1 and Fig. 2) need significant enhancement. The orientation (alignment) of both elementary cells (Al2O3 and Fe matrix as well) must be highlighted with respect to the AlO / Fe interface, including the crystallographic directions. Last but not least, all colors must be connected with chemical elements directly in the figures, not only inside the text or figure description and the interface should be marked too. In the current state, both AlO and Fe look like independent non-interacting structures. If the dashed line represents the size of the simulation cell, then this must be also highlighted directly in the figures. In summary, both figures are completely unacceptable for any scientific manuscript. In general, all figures look like a mess of lines only.

Line 149: The sentence: “Six layers were involved in plane modeling, and the 149 three layers in the bo􏰀om were fixed to save computational time.” This must be highlighted in Fig. 2.

Line 128: This sentence is not clear to me. “All atoms were 128 fully relaxed to calculate interface distances.

”.

Line 221: Bovia Materials Studio 8.0 doesn’t have any reference paper? It is strange when this software was used to get all results presented in this work but with no link to any paper or web.

Line 222: I am missing the reference to the work PBE functional.

Line 227: The convergence criteria of the self-consistent cycle is per entire cell or per atom? This information is missing in the text.

Line 227: The displacement term should be more clearly described. What does it mean?

Line 232: The simulation size description contains only two dimensions. There is no information about the length perpendicular to Figs. 2 or 3 - Z-axe. For this reason, the audience cannot make any imagination about periodicity in this direction - there is no information about mutual influence across the adjacent cells with respect to the periodic conditions. So, there are still two options. The first one is that the Z-length is two small and hydrogen creates an artificial infinite line here, which is very unphysical and unrealistic. The second, the Z-length is large enough to prevent mutual H-interactions across the periodic boundary conditions. This information should be included in the manuscript.

Line 240: Fig. 3 is almost identical to Fig. 2. There are only hydrogen positions included in the second one. This is wasting the manuscript space.

Line 260: Fig. 4 is extremely large.

I do not feel judge the way the authors incorporate the temperature effect.

In summary, I see what the authors are trying to say to the audience, but I have significant problems with their presentation. I had to read the manuscript several times to understand it clearly. Some important information is missing or unclearly described. Also, the graphic editing (figures) is unacceptable. For these reasons, I cannot recommend the manuscript for publication in the current form. The final decision is a major revision or new resubmission if the changes take longer than usual.

Reviewer 2 Report

Comments and Suggestions for Authors

In my opinion authors must modify/clarify the following issues:

#1) Page 1. Lines 31,33,37,40, 44…. Please avoid lumping references such as “[1-4]”. For the sake of clarity please describe each one individually in the main text.

#2) Page 1. Line 35. Please provide a reference for this statement

#3) Page 4. Figure 1. The box with dashed line has any meaning in this figure?.

#4) Page 5. Figure 2. The box with dashed line has any meaning in this figure?. In addition, please add a description of colors in the same way used in figures 1 and 3.

#5) Page 6. Equation 4. For the sake of clarity, please add a blank space between variables “RT” and between“”ln”. Please do the same for all equations.

#6) Page 6. Eq. 11. Please do not use “x” for scalar product

#7) Page 7. Line 237. Please do not use bold for Figure 3 in the main text.

#8) Page 7. Line 241. Figure 3. The box with dashed line has any meaning in this figure?. In addition, pleas add a description of the labels “1-4” in Figure caption.

#9) Page 8. Figure 4. The size of the x-axis and y-axis labels is very big, please reduce it

#10) Page 8. Lines 273-277. For the sake of clarity, please discuss these results deeply, explaining the reasons why.

#11) Additional descriptions of the Methods used for obtaining the results plotted in Figures 5-6, 9-11,13  are needed.

#12) Page 15. Figure 9. The label “a” and “b” are missed in the figure. Please, do the same for Figure 5 (Page 10).

Round 2

Reviewer 1 Report

Comments and Suggestions for Authors

I went through the revised version of the manuscript and I appreciate the changes made by the authors. Unfortunately, one issue still remains. 

The authors made changes to Fig. 1. However, the Fe matrix (e) and (f) still looks like chaotic mess due to too many lines, but bcc structure is simple and I hope that readers can create some good view. I appreciate that Fig. 1 contains details of simulation cells (a). In summary, Fig. 1 looks much better than in the previous version, however, the right bottom part contains a tiny coordinate system from the VESTA software (I suppose). It should be there or not? If yes, what Is the meaning of such small x ,y and z color coordinate system? Also, I cannot imagine how is oriented the vector a3. Fig. 1 (b) is extremely huge. I consider this as a minor issue.

previous question and answer:

 Line 149: The sentence: "Six layers were involved in plane modeling, and the three layers in the bottom were fixed to save computational time." This must be highlighted in Fig. 2. 

o As suggested, the sentences are revised as "to save computational time, the three layers in the bottom of the Al2O3 inclusion and two layers in the bottom of the Fe matrix were fixed, as indicated in Figure 2b. Moreover, three layers at the sides were also fixed (Figure 2c)" in p. 5 and 6. Figure 2 is revised to indicate the fixed and relaxed layers. 

Here, one important condition is not clear to me. I do not understand why fix the layers highlighted in Fig. 2 (c).  A justification of such fixing cannot by supported only by saving computational time only. Such model is very unusual and the authors should provide more details about it. Moreover, when I look to the simulation time then I see vacuum space along axe z. This means, that this model has at one side surface atoms relaxed and at the other side fixed. This is extremely confusing. Also the thickness along this direction is extremely small and the final structure has only a few layers. For this reason, my conclusion is still major revision until the authors justify their model.

previous question and answer:

 Line 128: This sentence is not clear to me. "All atoms were fully relaxed to calculate interface distances.". 

o After construction of the interface layer according to an existing model [46] with a similar Fe/Al-oxide system, the interface distances were calculated. The steps included: (i) Construct an interface layer model according to the existing interface data, (ii) Set all atoms in the model to be fully relaxed, (iii) Conduct geometry optimization to determine a stable interface configuration, and (iv) Measure the distance of the studied interface. 

o The descriptions are added in p. 3. 

The authors wrote: "Conduct geometry optimization to determine a stable interface configuration". This is not correct. Along the x and z direction is vacuum and no geometry optimization cannot be performed. One can optimize only ionic positions. In summary, such information is misleading as only along the x direction is structure periodic (no vacuum).

previous question and answer:

Line 260: Fig. 4 is extremely large. 

o Yes, the size of Figure 4 is properly reduced in the revised manuscript in p. 11. 

I still thing that the figure size is too much. But this must decide the journal office, not me. 

Summary:

In summary, all answers provided by the authors are okay except one issue. This issue is related to Fig. 2c and the model periodicity. Along this axe is the structure thin and fixing atoms only at one surface while the opposite one is relaxed (along the z directions) strange. The authors should provide a justification that this model is not influenced by any surface or very strange relax/unrelax ionic conditions. 

Round 3

Reviewer 1 Report

Comments and Suggestions for Authors

I went through the changes made by the authors.

The authors wrote:

The tiny coordinate system in the right bottom indicates the interaxial angles. The lattice parameters are marked in the figure. Such a coordinate system representing the interaxial angles follows the customary usage. The description is also in the caption of Fig. 1 in p. 5.

This I understand. I also use Vesta software for displaying structures. However, these axes are so small that they look like a forgotten anomaly from the creation process of the image. But this is the authors decision. In my opinion they look very strange.

Line 137: "Conduct geometry optimisation along Z direction to determine the interface distance" This is not geometry optimisation at all as there is a vacuum.

According to my understanding, geometry optimisation means changing the cell shape with respect to the stress tensor components, e.g. structure shape optimisation to reach zero or specific values of the stress tensor. But there is a vacuum, and any geometric optimisation along this direction may lead to decreasing the vacuum space or some unexpected results. So, I suppose that the authors created some vacuum at this direction and kept the cell shape. I recommend to correct/check this before the manuscript publication; otherwise, the audience may be very easily confused.

The authors response:

Regarding the model thickness and fixing atoms "It has been proposed that four layers in BCC Fe slab can make the surface energy consistent [31,55,56]. For α-Al2O3, it was verified...."

According to my opinion, such model is very small, oversimplified, and might not represent the real physical state. This model is so thick that it looks like the MoS2 2D structure and not some interface between two phases, unless we talk about nanoparticles. But at least the authors provided a better model description. The judgement of such model can make readers by themselves.

Author Response

Manuscript ID: cmd-2860061

I went through the changes made by the authors.

The authors wrote:

“The tiny coordinate system in the right bottom indicates the interaxial angles. The lattice parameters are marked in the figure. Such a coordinate system representing the interaxial angles follows the customary usage. The description is also in the caption of Fig. 1 in p. 5.”

This I understand. I also use Vesta software for displaying structures. However, these axes are so small that they look like a forgotten anomaly from the creation process of the image. But this is the authors decision. In my opinion they look very strange.

  • Thank you for the comment. Indeed, the tiny coordinate systems look strange in the figures. Thus, they are removed from Figures 1, 2, 8 and 10 in the revised manuscript.

Line 137: "Conduct geometry optimisation along Z direction to determine the interface distance" This is not geometry optimisation at all as there is a vacuum.

According to my understanding, geometry optimisation means changing the cell shape with respect to the stress tensor components, e.g. structure shape optimisation to reach zero or specific values of the stress tensor. But there is a vacuum, and any geometric optimisation along this direction may lead to decreasing the vacuum space or some unexpected results. So, I suppose that the authors created some vacuum at this direction and kept the cell shape. I recommend to correct/check this before the manuscript publication; otherwise, the audience may be very easily confused.

  • Yes, the vacuum slab was added in this study, and a supercell was created before geometry optimization. The comment is added for clarification in p. 3.

The authors response:

Regarding the model thickness and fixing atoms "It has been proposed that four layers in BCC Fe slab can make the surface energy consistent [31,55,56]. For α-Al2O3, it was verified...."

According to my opinion, such model is very small, oversimplified, and might not represent the real physical state. This model is so thick that it looks like the MoS2 2D structure and not some interface between two phases, unless we talk about nanoparticles. But at least the authors provided a better model description. The judgement of such model can make readers by themselves.

  • Many thanks for the comments. This study investigates the hydrogen adsorption at the Al2O3-ferrite interface. As a common inclusion within pipeline steels, Al2O3 inclusions are normally several microns in diameter. This is beyond the computational capability. A rectilinear model is thus developed to represent the interface. Relevant descriptions are included in the Introduction section.